# Intragland Expression of the Shh Gene Alleviates Irradiation-Induced Salivary Gland Injury through Microvessel Protection and the Regulation of Oxidative Stress

**DOI:** 10.3390/antiox13080904

**Published:** 2024-07-26

**Authors:** Meijun Hu, Liang Hu, Tao Yang, Bowen Zhou, Xuanhe Feng, Zhipeng Fan, Zhaochen Shan

**Affiliations:** 1Laboratory of Molecular Signaling and Stem Cells Therapy, Beijing Key Laboratory of Tooth Regeneration and Function Reconstruction, School of Stomatology, Capital Medical University, Beijing 100050, China; humeijun_1110@163.com; 2Outpatient Department of Oral and Maxillofacial Surgery, School of Stomatology, Capital Medical University, Beijing 100050, China; huliang@mail.ccmu.edu.cn (L.H.);

**Keywords:** ionizing radiation, Hedgehog pathway, salivary gland (SG) injury, oxidative stress, microvessel

## Abstract

Radiation-induced salivary gland injury (RISGI) is a common complication of radiotherapy in patients with head and neck cancer. Intragland expression of the Sonic Hedgehog (Shh) gene may partially rescue irradiation (IR)-induced hyposalivation by preserving salivary stem/progenitor cells and parasympathetic innervation, maintaining resident macrophages, and maintaining microvascular density. Previous studies have revealed that Ad-Rat Shh transduction through the salivary glands of miniature pigs can ameliorate oxidative stress-induced microvascular dysfunction after radiotherapy. Changes in the parotid salivary flow rate were analyzed, and the parotid tissue was collected at 5 and 20 weeks after IR. Changes in the Hedgehog pathway and vascular function-related markers (vascular endothelial growth factor (VEGF) and CD31) and oxidative stress-related markers were detected via immunohistochemistry, immunofluorescence, and Western blotting. A stable Shh-overexpressing cell line was generated from human umbilical vein endothelial cells (HUVECs) and exposed to 10 Gy X-ray irradiation, after which endothelial cell proliferation, senescence, apoptosis, and vascular function were evaluated. We found that intragland expression of the Shh gene efficiently alleviated IR-induced parotid gland injury in a miniature pig model. Our results indicate that the antioxidative stress and microvascular-protective effects of the Hh pathway are regulated by nuclear factor-erythroid 2-related factor 2 (Nrf2).

## 1. Introduction

Head and neck cancer (HNC) was the ninth most common cancer in the United States in 2021 [1]. A total of 50,100 new cases and 10,850 deaths from head and neck cancer were reported. Standard treatments (surgery, radiotherapy, and chemotherapy) were selected according to the stage of the HNC. As an essential component of multimodal treatment for the majority of locally advanced head and neck cancers, radiotherapy can be combined with surgery, chemotherapy, targeted therapy, and hyperthermia to significantly improve the local control rate and prognosis of tumors [2]. However, radiotherapy for HNC often leads to acute and chronic complications, such as mucositis, dysphagia, hoarseness, xerostomia, radioactive osteonecrosis, subcutaneous fibrosis, lockjaw, taste loss, thyroid dysfunction, esophageal stenosis, and hoarseness [3].

Xerostomia is the most common and prominent complication during and after radiotherapy for HNC patients [4]. Due to the high sensitivity of the salivary gland (SG) to radiation (especially serous acinar cells with secretory functions) and its superficial anatomical location, damage to the SG is usually irreversible when the cumulative total dose exceeds 50 Gy [5]. A decrease in salivary secretion directly or indirectly affects the normal physiological functions of the mouth, such as speech, chewing, and swallowing. Moreover, a shift in the oral microbial population towards cariogenic bacteria, a decrease in salivary flow, and changes in salivary composition (buffering capacity, pH value, and immunoglobulin concentration) may lead to the rapid progression of rampant radiation caries [6].

RISGI is a global form of tissue dysfunction involving multiple tissues and cellular processes, including water channel injury, microvascular endothelial cell injury, parasympathetic nerve damage, and blocked calcium ion signaling [7,8]. There is considerable evidence that IR targets microvascular endothelial cells [9]; for example, the endothelium is recognized as a major target of irradiation-induced lung and brain injury [8,10]. The endothelial survival factors vascular endothelial growth factor (VEGF), acidic and basic fibroblast growth factor (aFGF, bFGF), and interleukin [11] protect the gut from radiation damage. Oxidative stress (OS) refers to the imbalance between oxidation and antioxidation in the body, leading to inflammatory neutrophil infiltration, increased protease secretion, and the production of many oxidative intermediates. Currently, OS is considered an important factor leading to RISGI [7,11]. Thus, alleviating the increase in oxidative stress and reducing microvascular injury induced by IR in the SG play important roles in maintaining the biofunction of the SG.

Current clinical treatments for RISGI are mainly symptomatic and include drug therapy (pilocarpine [12]), alternative therapy (dry mouth glue, artificial saliva), and hyperbaric oxygen therapy. In addition, stem cell therapy [13,14], oral stents, temporary submandibular gland transplantation [15], and gene therapy can also alleviate symptoms, such as dry mouth, caused by radiation damage to the salivary gland to some extent, but they cannot fundamentally meet the needs of patients.

Shh, a typical morphogenetic hormone, is an important secretory signaling molecule in developing tissues that regulates epithelial–mesenchymal transition during embryonic development. Some previous studies have shown that transient activation of the Hedgehog pathway can rescue IR-induced hyposalivation by promoting DNA repair, alleviating oxidative stress, suppressing salivary gland cell senescence [11], protecting salivary stem/progenitor cells [16] and parasympathetic innervation and maintaining resident macrophages [17]. Moreover, intragland Shh gene delivery may partially maintain microvascular density [17]. However, the mechanism by which activation of the Hh signaling pathway affects vascular endothelial cells after radiotherapy and maintains oxidative stress homeostasis in the salivary glands of miniature pigs after irradiation is still unclear.

Oxidative stress and its associated injuries, including microvascular circulation disorders, play a vital role in driving salivary gland injury following radiotherapy. In this study, miniature pigs were used as experimental animals to investigate the protective effect of Shh gene transfer on salivary glands after radiotherapy by alleviating oxidative stress and protecting microvessels. We examined the role of Nrf2 in Hh pathway-dependent antioxidative stress in vitro.

## 2. Materials and Methods

### 2.1. Animal Experiments

Eight-month-old male BAMA miniature pigs (weighing approximately 25–35 kg) were purchased from the Institute of Animal Science of the Chinese Agriculture University (Beijing, China). All the animals were housed in cages in a clean environment at the appropriate temperature, humidity, water, and food.

In accordance with our previous study [17], cranial CT images of the animals were obtained to delineate the central target area of the parotid gland, and one week later, the animals were again imaged via cranial CT to delineate the central target area of the parotid gland. After one additional week, CT-guided radiotherapy was administered via the Elekta Synergy accelerator (Elekta, Stockholm, Sweden) as the radiation source (the average energy of the radiation was 6 Mv, and the irradiation distance was 1 m); a dose of 20 Gy was delivered to the unilateral parotid gland over three consecutive days at a rate of 3.2 Gy/min. Retrograde delivery of Ad-Rat-Shh or Ad-Rat-GFP via the parotid ducts was performed 4 weeks after radiotherapy.

### 2.2. Salivary Flow Rate Detection

Saliva flow rates were measured every two weeks. After anesthesia, the miniature pigs were placed in a supine position on a special scaffold. Ten minutes before the experiment, a dose of 0.1 mg/kg pilocarpine was injected behind both ears. Approximately 10–20 min after injection, when a pool of saliva formed in the mouth of the miniature pigs, an inner Carlins–Crittenden ring connected to a negative pressure aspirator was placed at the nipple of the right parotid duct of each experimental animal. The outer ring was connected to an outflow channel to prevent mucosal swelling, which could lead to duct obstruction. The first drop was discarded until the second drop of cool liquid appeared, and then the saliva sample was collected and kept on ice for 10 min to record the volume of secretion. 

### 2.3. Cell Culture

Human umbilical vein endothelial cells (HUVECs) and VascuLife medium were purchased from Lifeline Cell Technology (FC-0003 and 08837, San Diego, CA, USA). HUVECs have characteristic features of vascular endothelial cells, including a rounded or polygonal shape, distinct cell borders, and multiple cellular prominences. The Shh overexpressing lentivirus and control GFP virus were purchased from Suzhou Gemma Genetics Co., Suzhou, China. The cells were pretreated with ML385 (0.25 µmol/L, MedChemExpress, Monmouth Junction, NJ, USA) for 2 h prior to IR. HUVECs were irradiated with a single dose of 10 Gy using an Elekta Synergy (Trilogy Varian, Palo Alto, CA, USA) at a rate of 1.48 Gy/min.

### 2.4. Histology, Immunohistochemistry (IHC), and Immunofluorescence (IF) Analyses

The animals were sacrificed at 5 and 20 weeks after radiotherapy (1 and 16 weeks after gene therapy). Parotid tissues were harvested, and the glandular follicles were peeled off, fixed in 4% paraformaldehyde, dehydrated in graded alcohol solutions, embedded in conventional paraffin, and sectioned at 5 μm. The sections were stained with hematoxylin and eosin and examined for morphological changes. Masson’s trichrome staining was used to compare the degree of fibrosis in the parotid tissue after radiotherapy, and Periodic Acid-Schiff stain was used to compare the level of glycogen in the parotid follicles in pigs from each group. For immunostaining, the sections were blocked with 10% donkey serum for 1 h at room temperature. The colocalization of CD31 (1:100, Bioss, Beijing, China) and Ptch1 (1:200, Abcam, Cambridge, MA, USA) was detected by IF staining, in which the sections were incubated with a fluorescent secondary antibody to the corresponding species of primary antibody, followed by nuclear staining with DAPI (F6057, Sigma, St. Louis, MO, USA). IHC was used to determine the expression of VEGF (1:100, Bioss bs1313R), CD31 (1:300, Bioss), p-H2AX (1:200, CST 80312), Nrf2 (1:200, Proteintech 16396-1), and NOX4 (1:50, Proteintech 14347-3). The secondary antibody for IHC was conjugated to HRP (1:10,000, Abcam), DAB was used as the chromogen, and the sections were counterstained with hematoxylin. The positive parts are marked with red triangles. The cells were fixed in paraformaldehyde and treated with 0.1% Triton X-100 for 5 min. The dilution ratios of the primary antibodies used were as follows: VEGF (1:100, Bioss), Gli1 (1:200, Proteintech, Rosemont, IL, USA), Bax (1:100, Proteintech), and Bcl-2 (1:100, Proteintech). All images were taken with a microscope (BX61, Olympus, Tokyo, Japan). At least 3 different sections were imaged, and 3 different fields per section were included in each group.

### 2.5. Western Blot

Protein lysates (RIPA:PMSF:PIC = 100:1:1) were prepared, and total cellular proteins were extracted on ice. The protein concentration was determined by the Bradford method (Bio-Rad Laboratories, Hercules, CA, USA). Protein fractions were separated by sodium dodecyl sulfate–polyacrylamide gel electrophoresis (SDS–PAGE) and transferred to PVDF membranes. The membranes were incubated overnight at 4 °C with the following primary antibodies: Shh (1:1000, ABclonal A7726, Wuhan, China), Gli1 (1:5000 Proteintech), VEGF (1:300, Bioss BS-1313R), CD31 (1:1000, Abcam ab281583), Bcl-2 (1:2000, Abcam), Bax (1:1000, Abcam), p53 (1:1000, Proteintech), Nrf2 (1:1000, Proteintech 16396-1), p-H2AX (1:1000, CST 80312), and β-actin (1:50,000, ABclonal). The Western blot analysis was repeated three times, and the proteins were quantified using Image J1 software.

### 2.6. Cell Proliferation, Apoptosis, and Senescence Assays

Cell proliferation was assessed using a Cell-Counting Kit-8 (CCK-8) assay (DOJINDO, Kumamoto, Japan). HUVECs at the logarithmic growth stage were seeded into 96-well plates at a density of 4000 cells per well, with 3 replicates per treatment group. Radiation treatment was administered 24 h after cell plating. The extent of parotid gland follicular cell apoptosis was determined by TUNEL staining (Beyotime) performed according to the manufacturer’s instructions. Images were captured with a fluorescence microscope at 400× magnification. Three fields from each group were randomly selected, and the ratio of the number of TUNEL-positive cells in each field to the total number of cells in each field was calculated as the apoptosis rate. Senescence-associated β-galactosidase (SA-β-gal) staining was performed using an SA-β-gal staining kit (Beyotime).

### 2.7. Quantitative RT-PCR

Quantitative RT-PCR (qRT-PCR) was conducted in accordance with the previously published protocol [11]. Primers for GAPDH and GDF15 were designed using the Primer-BLAST (https://www.ncbi.nlm.nih.gov/, accessed on 2 July 2024).

### 2.8. Statistical Analysis

All the data were analyzed with Prism 8 software (version 8.0.2, GraphPad, San Diego, CA, USA) using one-way ANOVA with Tukey’s multiple comparison test and Student’s unpaired *t*-tests to evaluate the differences between groups. Graphs were generated using Prism 8. Statistical significance was set at *p* < 0.05.

## 3. Results

### 3.1. Salivary Gland Transduction with Ad-Rat-Shh Effectively Alleviates IR-Induced Hyposalivation in Miniature Pigs

A significant sign of RISGI is a reduced salivary flow rate. The salivary flow rate of the miniature pigs was measured every 2 weeks before and after radiotherapy. Analysis of adenovirus transduction through the parotid duct at 4 weeks after radiotherapy (Figure 1A) revealed that radiotherapy damaged the salivary glands and led to a continuous decrease in the salivary flow rate, whereas transduction of the salivary gland with Ad-Rat Shh partially restored the salivary flow rate. The salivary flow rate in the advanced IR and IR+GFP groups was only 20% of that in the normal group, and that in the Shh gene therapy group was 40% of that in the nontransduced group. (Figure 1B). The weight of the parotid gland decreased significantly after IR, and only 40% of the normal weight was maintained. In the IR+Shh group, 60% of the glandular weight was maintained (Figure 1C). H&E (Figure 1D) and Masson’s trichrome staining showed that 20 weeks after radiotherapy, the normal structure of the salivary glands in the IR and IR+GFP groups was severely damaged, and the acini, which had disappeared completely or appeared atrophied, vacuolated, and dilated in the catheter were replaced by a large amount of fibrous tissue (Figure 1E,F). Tissue fibrosis is the characteristic manifestation of post-radiation injury, and fibrosis of ducts and the perivasculature leads to disturbances in salivary gland secretion. The lobular structure of the glands in the Shh gene transduction group was clear, the acinar structure was more complete, zymogen particles were abundant (Figure 1G,H), and the degree of fibrosis was significantly lower than that in the other radiotherapy groups (** *p* < 0.01).

In the initial stage after radiotherapy, acinar cells undergo apoptosis, which is subsequently accompanied by tissue dysfunction. TdT-mediated dUTP nick-end labeling was used to detect DNA breaks in the nucleus. In this study, we observed that at 20 weeks after IR, the proportion of apoptotic cells per unit field in the IR and IR+GFP groups was markedly greater than that observed in the NT group and gene therapy group (Figure 1I,J). Therefore, the Hh signaling pathway, which is activated by transduced genes in the parotid glands of miniature pigs, effectively reduced the degree of apoptosis in acini. 

### 3.2. Intragland Shh Gene Delivery Increases Microvascular Density by Promoting the Expression of VEGF and CD31

Shh is an indirect angiogenic factor that upregulates numerous angiogenic cytokines. Shh is also involved in embryogenesis and angiogenesis after ischemic injury. Fluorescence colocalization revealed high expression of PTCH1 in CD31-positive cells, suggesting that Shh was highly expressed in vascular endothelial cells after retrograde delivery of the Shh gene into the parotid duct of miniature pigs (Figure 2A). Compared with those in the NT group, the expression of CD31 and Ptch1 was significantly lower in the IR and IR+GFP groups after IR, while Ptch1 was highly expressed in the IR+Shh group, and the microvascular density and CD31 expression intensity were also significantly greater in the IR+Shh group than in the control group. Immunohistochemical staining revealed that Shh gene transduction significantly increased the expression of CD31 and VEGF in the parotid gland after IR. Immunohistochemical detection of VEGF revealed that radiotherapy significantly reduced VEGF expression and microvascular density in the parotid tissue and that the microvascular density of the IR+Shh group was restored to the control level (Figure 2B–E). Therefore, we can conclude that delivery of Ad-Rat-Shh promotes the expression of VEGF and CD31 to increase microvascular synthesis and protect salivary gland function. Blood vessel formation is associated with increased salivary secretion, suggesting that transduction of Shh genes may promote functional recovery of the glands by upregulating VEGF expression.

### 3.3. Activation of the Hh Signaling Pathway Regulates Oxidative Stress in Salivary Gland Tissues

Oxidative stress represents a significant early change following radiotherapy, frequently resulting in DNA double-strand breaks (DSBs). The phosphorylation of histone H2AX (p-H2AX) serves as a crucial marker of DSBs. The levels of p-H2AX were significantly increased after radiotherapy, which suggests that oxidative stress-induced damage was initiated in the gland. Following Ad-Shh gene therapy, the Hh signaling pathway was activated, p-H2AX levels were significantly reduced, and oxidative stress was alleviated (Figure 3A,B). Nrf2 and NOX4 are important proteins in the body that regulate oxidative stress levels. IHC staining showed that IR significantly downregulated Nrf2 levels and increased NOX4 levels. Activation of the Hh signaling pathway may regulate the level of oxidative stress through upregulation of Nrf2 and downregulation of NOX4, thereby playing a role in reducing oxidative stress-induced damage (Figure 3C–F). Subsequent Western blot analyses further verified that after Ad-Shh transduction, Nrf2 and NOX4 may be regulated to effectively reduce oxidative stress-induced damage to the parotid glands of miniature pigs (Figure 3G,H).

### 3.4. Overexpression of the Shh Gene Activates the Hh Signaling Pathway and Promotes Vascular Function in Endothelial Cells

Microvasculature is an important component of the salivary gland and changes markedly in the early stage of radiation injury. To investigate the possible mechanism by which the Shh gene may prevent and treat salivary gland microvascular injury, we established HUVECs from the human umbilical vein that stably overexpress Shh, and we treated them with 10 Gy of radiation. The protein expression level of Shh was detected via Western blot (Figure 4A,B). The CCK-8 results showed a reduction in the proliferative activity of vascular endothelial cells after IR. Compared with the effects of IR or IR+GFP alone, the overexpression of Shh at 24 h and 48 h after IR was observed to maintain the proliferative activity of endothelial cells (* *p* < 0.05). However, there was no significant difference in cell proliferation among the groups at 4 h after IR (Figure 4C). At 4 h, 24 h, and 48 h after radiotherapy, the protein expression of Shh and Gli1 in HUVECs was down-regulated, and the Hh signaling pathway was inhibited. Conversely, the expression of Shh signaling molecules and Gli1 downstream of the Hh pathway was significantly upregulated by Shh gene transduction (Figure 4D,E). VEGF is a specific mitogen of vascular endothelial cells that can promote the proliferation of vascular endothelial cells, increase the permeability of blood vessels, promote changes in the morphology and cytoskeleton of vascular endothelial cells, and stimulate their migration and growth. Consequently, VEGF plays an important role in the regulation of angiogenesis. At 4 h and 24 h after IR, IF analysis revealed that the expression level of intracytoplasmic VEGF in the IR+shh group was significantly greater than that in the IR and IR+NC groups (Figure 4F). Compared with that in the control group (Blank), the expression of CD31 in the IR and IR+NC groups was significantly lower, while the overexpression of Shh promoted VEGF and CD31 expression, thereby reversing the IR-induced damage to the vasogenic function of HUVECs (Figure 4G–I). This effect was statistically significant (*p* < 0.05).

### 3.5. Activation of the Hh Signaling Pathway Protects Endothelial Cells by Regulating the Apoptosis Signaling Pathway

The primary mechanism of cell death induced by IR is apoptosis. Immunofluorescence and Western blot analysis demonstrated that the expression levels of p53 and Bax were elevated in the IR and IR+NC groups compared with those in the blank group (Figure 5A,B,E,F). Conversely, the expression of the antiapoptotic factor Bcl-2 was significantly reduced (Figure 5A,B). The overexpression of Shh was observed to reduce endothelial cell apoptosis by upregulating Bcl-2 and downregulating p53 and Bax. The results of β-gal staining and quantitative analysis revealed that the number of β-gal-positive cells in the IR+Shh group was significantly lower than in the IR and IR+NC groups (Figure 5C,D, *n* = 3, ** *p* < 0.01, * *p* < 0.05). Subsequently, we found that radiotherapy significantly induced cell senescence, resulting in upregulated protein expression levels of p21 and p16, which are related to cell senescence. However, overexpression of shh could effectively inhibit cell senescence (Figure 5E). These findings indicate that activation of the Hh signaling pathway may mitigate the detrimental effects of ionizing radiation on endothelial cells by modulating the apoptosis signaling pathway.

### 3.6. The Overexpression of Shh Activates the Nrf2 Signaling Pathway in Irradiated Endothelial Cells

Growth differentiation factor-15 (GDF15) is upregulated in various tissues and cells in response to ionizing radiation and is a critical factor in the induction of reactive oxygen species (ROS) and cellular aging in human endothelial cells [11]. Studies have shown that transient Hh activation reduced oxidative stress caused by IR through inhibition of GDF15 upregulation [11]. In the present study, we have demonstrated that the overexpression of the Shh gene in HUVECs has the effect of mitigating the up-regulation of GDF15 mRNA expression that occurs in response to IR (Figure 6A). Intracellular DNA double-strand breaks occur within minutes of exposure to ionizing radiation. When DNA damage occurs in cells, histone H2AX is phosphorylated, and the resulting product p-H2AX may be used as a biomarker for detecting DSBs. p-H2AX expression was increased, and Nrf2 expression was decreased in HUVEC after 10 Gy radiation treatment. Conversely, overexpression of Shh inhibited p-H2AX production and maintained Nrf2 levels at preradiotherapy levels. (Figure 6B). Consequently, we hypothesized that activation of the Hh pathway may influence the radiation sensitivity of cells by regulating the DNA damage repair process through Nrf2. To ascertain whether the protective effect of Shh overexpression on endothelial cells is mediated by the Nrf2 signaling pathway, endothelial cells overexpressing the Shh gene were treated with the specific Nrf2 inhibitor ML385 for a period of 2 h prior to radiotherapy, and the Nrf2 level was subsequently measured after IR. The results demonstrated that the capacity of Shh overexpression to increase Nrf2 expression was markedly diminished by ML385 (* *p* < 0.05) (Figure 6C,D). However, ML385 did not influence the level of Shh after IR (Figure 6C,E). DNA damage and apoptosis were detected, and ML385 pretreatment counteracted the protective effect of Shh overexpression on endothelial cells, increasing the levels of p-H2AX and Bax and decreasing the level of Bcl-2 (* *p* < 0.05, Figure 6F,G,K). In addition, the expression of VEGF and CD31 was significantly downregulated in HUVECs after IR (Figure 6H–J). The overexpression of Shh partially restored the vascular function of endothelial cells, but this effect was blocked by ML385.

In conclusion, our results suggest that activation of the Hh pathway can reduce oxidative stress, inhibit cell senescence and apoptosis, increase microvascular density by promoting vascular function, and prevent IR-induced salivary gland injury by regulating Nrf2 (Figure 7).

## 4. Discussion

As an important part of sequential therapy for head and neck cancer, radiotherapy may cause damage to the normal tissue structure of the target area while eliminating tumor cells [18]. Given the physiological characteristics and anatomical location of the salivary glands, they are susceptible to damage from radiation, which can ultimately result in a range of dysfunctions that affect the quality of life and prognosis of patients. The primary mechanism underlying salivary gland damage induced by radiotherapy is the loss of functional acinar cells. The salivary glands exhibit a delayed response to radiation, manifesting initially as apoptosis, particle leakage, and acinar cell lysis. Late injury is characterized by selective membrane damage, which results in cell dysfunction. Moreover, classical killer cells and damage to the cell microenvironment prevent acinar progenitor cells from differentiating into mature acinar cells. Therefore, recent studies suggest that microvascular injury may be the main cause of late salivary gland dysfunction [10].

High expression of the Shh gene in the mouse submandibular gland after ligation and release may be related to self-repair after injury. Shh, a typical morphogenetic hormone, regulates the epithelial–mesenchymal transition during embryonic development. The Hh signaling pathway promotes the proliferation, migration, and survival of endothelial cells and plays a crucial role in the normal development of the vasculature [19]. The Hh pathway plays an important role in the treatment of many diseases, such as inducing and promoting angiogenesis repair after stroke in rats [20]. The Shh gene accelerates diabetic wound healing by enhancing endothelial progenitor cell-mediated microvascular modeling [21]. The angiogenic potential of Wharton’s jelly mesenchymal stem cells (WJ-MSCs) is increased after Shh expression [22]. Moreover, Shh can act as an indirect angiogenic factor to regulate the expression of VEGF-1, Ang-1, and Ang-2 and has potential therapeutic use in ischemic diseases [23]. VEGF plays an important role in the vascularization and development of various organs, as it promotes the regeneration of skin, bone, heart, cornea, and other tissues. Hedgehog signaling plays a key role in endothelial cell viability and apoptosis, and previous studies have shown that silencing the key transmembrane conductance regulator of the Hh signaling pathway significantly reduces the upregulation of ANG-1, VEGFA, VEGFR2, and ENOS expression, suggesting that angiogenesis-related genes may be located downstream of the Hedgehog-Gli1 signaling pathway and may be regulated by this pathway. Some researchers have shown that IR, endothelial cell integrity, and SG function are closely related: SG microvascular endothelial cells may be the early target of single-dose IR, and salivary gland dysfunction after IR can be prevented by local application of Ad-VEGF [10]. In experimental retinal neovascularization, inhibition of the Hh pathway leads to decreased levels of VEGF and the classical Shh target PTCH1, which indicates that the Hh pathway functions upstream of VEGF [24,25]. Consistent with the above studies, we also confirmed that Shh can mediate VEGF to induce angiogenesis [25]. CD31 is an endothelial cell marker that is commonly used to assess microvessel density [10,17], and in vivo experiments have shown that Ptch1 and CD31 are coexpressed in blood vessels. Shh is an important vector of angiogenesis that influences endothelial cell behavior, promotes blood vessel formation, and coordinates the pattern and maturation of blood vessels. Interactions between Shh and other signaling pathways, such as VEGF, further fine-tune the vascular development process [26]. Therefore, we can conclude that transcatheter transduction of Ad-Rat Shh after radiotherapy promotes the expression of VEGF and CD31 to increase microvessel formation and protect the function of salivary glands by activating the Hh signaling pathway. The formation of blood vessels is associated with increased salivary secretion, which suggests that transducing the Shh gene may promote the recovery of functional glands after IR by upregulating VEGF expression.

Radiotherapy can directly induce the body to produce many reactive oxygen species [27], which results in oxidative stress injury [28], neutrophil inflammatory infiltration, and increased protease secretion, leading to normal tissue and cell damage [29]. Various studies have shown that oxidative stress is associated with the pathogenesis of many other diseases, such as cancer [30], age-related Alzheimer’s disease [31], Parkinson’s disease [32], microvascular and macrovascular complications in patients with diabetes [33], and cardiovascular events in patients with chronic kidney disease (CKD). NOX4 is an important member of the NADPH oxidase family that acts as part of the electron respiratory chain, which transfers electrons from NADPH to molecular oxygen to produce ROS [34,35]. Our results suggest that NOX4, which is elevated in salivary gland tissue, can be significantly reduced by activating the Hh signaling pathway. Ionizing radiation and oxidative stress can cause DNA damage and apoptosis [36,37]. p-H2AX is an important marker of DNA double-strand breaks [38,39]. In several studies, high expression of p-H2AX was strongly associated with malignant cancer characteristics and poor patient survival. Therefore, p-H2AX may also be an independent prognostic predictor of head and neck cancer and a potential therapeutic target. Nrf2 is an important multifunctional cellular transcription factor that plays a central regulatory role in cellular defense mechanisms. Since Nrf2 has a strong antioxidant regulatory ability and can promote radiation-induced DNA damage repair, increased expression of Nrf2 is generally considered to be an important cause of radiation resistance in cells [40]. By inducing and regulating the constitutive and inducible expression of a series of antioxidant proteins, Nrf2 can reduce cell damage caused by reactive oxygen species and electrophiles, maintain a stable state, and maintain the body’s dynamic redox balance, all of which are crucial for maintaining cell homeostasis. A previous study [11] has shown that retrograde delivery of the Ad-Rat-Shh gene via the submandibular duct can temporarily activate the Hh pathway in the mouse submandibular gland and protect against salivary gland dysfunction caused by radiation. In irradiated salivary glands and endothelial cells subjected to Shh gene transduction, p-H2AX expression decreased, whereas Nrf2 levels significantly increased, which indicates that activation of the Hh pathway can inhibit IR-induced DSBs and alleviate DNA damage.

Endothelial cells are the most sensitive cell type in the blood vessel wall to ionizing radiation, as radiation above 10 Gy can cause severe endothelial cell dysfunction and apoptosis [41]. Ionizing radiation causes DNA breakage in endothelial cells, and cell aging results in slow cell growth, which manifests as increased vascular permeability in vivo. Radiation-induced DNA damage also leads to the activation of the ataxia telangiectasia mutated gene (ATM) and DNA-dependent protein kinases, which phosphorylate p53 and trigger cell cycle arrest [42]. Bcl-2 is an oncoprotein found in cancer proteins in the nuclear membrane, some mitochondria, and the outer membrane that can protect cells from apoptosis. p53 can be used as a proapoptotic factor to guide the transcription and translation of the Bcl-2 family. Bax is an apoptosis regulatory gene that forms a heterodimer with Bcl-2 in vivo, and high expression of Bcl-2 inhibits apoptosis. In one study, when Bax was highly expressed, a homologous dimer was formed, and apoptosis was promoted. The overexpression of Shh inhibited the expression of SA-β gal and p53. Shh gene overexpression upregulated Bcl-2 and downregulated Bax expression after IR. However, when endothelial cells overexpressing Shh were treated with the Nrf2-specific inhibitor ML385 prior to IR, this protective effect disappeared, which suggests that Nrf2 may be located downstream of Shh. These findings suggest that Hh pathway activation can protect endothelial cells from radiation damage by activating Nrf2, alleviating DNA damage and oxidative stress, and inhibiting apoptosis through the p53 pathway.

To evaluate the safety of Hh pathway activation, Fiaschi and others constitutively activated the Hh signaling pathway in salivary glands for 15 weeks by transducing the Gli1 gene and reported that the glands were partially hyperplastic [43]. However, when the stimulus was removed, histological examination revealed that the glands had essentially returned to normal. Squamous cell carcinoma is the most common type of head and neck cancer. Hai et al. transfected the Shh gene into mouse SCC VII cells, the mouse submandibular gland, and the subcutaneous SCC tumors growing in nude mice; the downstream target genes rShh and Hh were significantly upregulated in the first two groups, but target gene expression had no significant effect on tumor growth or tumor growth after radiotherapy [16]. This finding suggests that local activation of the Hh pathway can be achieved in the mouse submandibular gland but that this pathway does not promote the growth of existing tumors outside the salivary gland. In addition, when adenovirus is used as a gene delivery vehicle, the expression of the target gene is transient, which may, to some extent, prevent the potential tumor-promoting effect of Shh gene expression; thus, its biological effect would be safer and more efficient. However, there are differences among rodents, large animals, and humans in terms of anatomical structure and biological function, and further verification of the safety of this therapy is still needed. Although gene therapy has certain research values and application prospects for alleviating radiation-induced dysfunction of the salivary gland [44], its application still requires further exploration to achieve better therapeutic effects.

RISGI is the consequence of a complex interplay of factors, including oxidative stress, apoptosis, aging, nerve damage, immune dysfunction, and microvascular circulation disorders. Among these, oxidative stress is the most important early insult that initiates subsequent microvascular circulation disorders. The results of this study suggest that activation of the Hh pathway can reduce the overall level of oxidative stress in the parotid glands of miniature pigs and ameliorate microvascular damage. The in vitro experiments demonstrated that the Hh pathway can activate Nrf-VEGF and thereby regulate microvascular function. However, current studies on the protection of transient Hh pathway activation on microvascular function are limited, and in vitro experiments can only show that Shh overexpression is beneficial for vascular function but not its effect on angiogenesis. Further experiments are needed to address this blind spot in the future.

## 5. Conclusions

Our findings suggest that activation of the Hh pathway can reduce oxidative stress, increase microvessel density by promoting vascular function, and prevent IR-induced salivary gland injury by regulating Nrf2. These data indicate that Shh gene therapy represents a feasible, innovative, and safe approach to clinically protect against salivary gland radiation injury. 

## Figures and Tables

**Figure 1 antioxidants-13-00904-f001:**
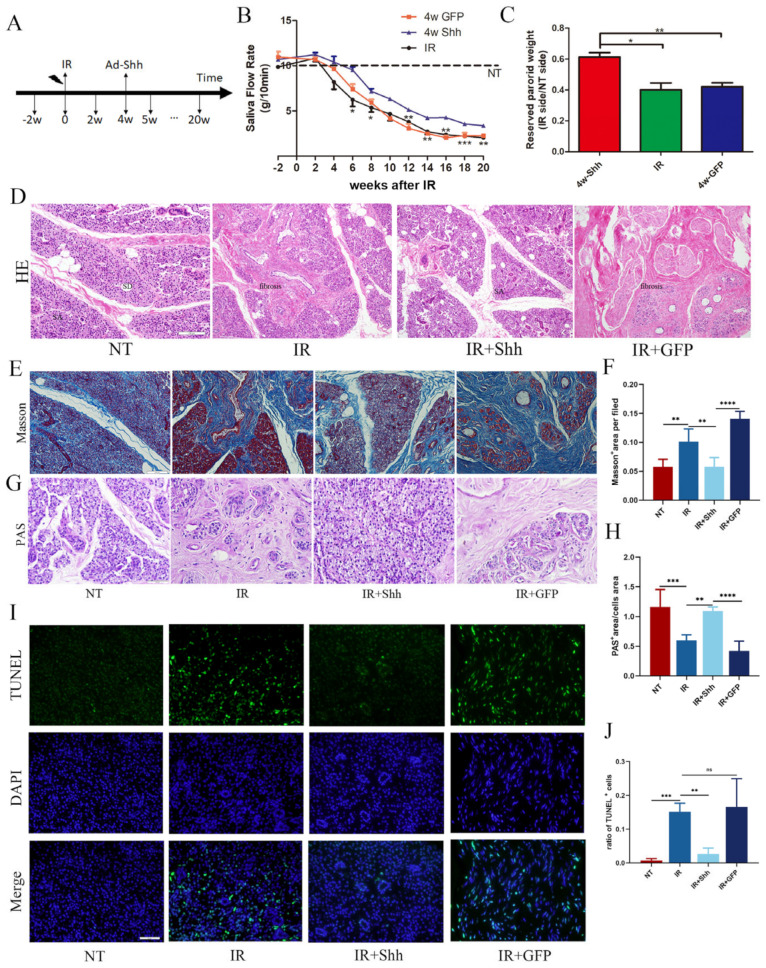
Saliva flow rate and histomorphological changes in irradiated parotid glands. (**A**) A flow chart of the radiotherapy and gene therapy process. Adenoviruses carrying the rat Shh gene or the GFP gene were delivered 4 weeks after irradiation, and saliva was collected every 2 weeks. At 5 and 20 weeks, the parotid tissue was obtained from the miniature pigs through sacrifice. (**B**) Stimulated salivary flow rate (g/10 min) of the parotid gland in the NT, IR, IR+Shh, and IR+GFP groups, *n* = 3. (**C**) After 20 weeks of radiotherapy, the remaining weight of the parotid gland in the IR group was found to be less than 40%, which could be sustained at 60% after gene therapy, *n* = 3. (**D**) H&E staining revealed the disappearance of the acinar structure and the proliferation of fibrous tissue in the parotid gland after irradiation. In contrast, the acinar area and zymogen particles of the IR+Shh group were notably abundant. SA: serous acinus; SD: secretory duct. (**E**,**F**) Masson’s staining revealed a significantly greater ratio of the fibrotic area in the IR and IR+GFP groups than in the NT group. However, gene therapy effectively alleviated the degree of fibrosis. (**G**,**H**) Pioneer tissues collected at week 20 were examined via PAS staining. Compared with those in the NT group, the acinar area and glycogen level were reduced in the IR and IR+GFP groups, and many functional acinus were preserved by Shh gene therapy. The positive acinar area was calculated at 400× magnification. (**I**,**J**) Green indicates TUNEL-positive expression, whereas blue indicates the nucleus. Radiotherapy induced apoptosis in various cells within the parotid gland tissue and the ratio of apoptotic cells was significantly reduced in the IR+Shh group. Scale bar, 50 μm; *n* = 3. The data are shown as the mean ± standard error of the mean (SEM); ns (not significant), * (*p* < 0.05), ** (*p* < 0.01), *** (*p* < 0.001), and **** (*p* < 0.0001) indicate significant differences.

**Figure 2 antioxidants-13-00904-f002:**
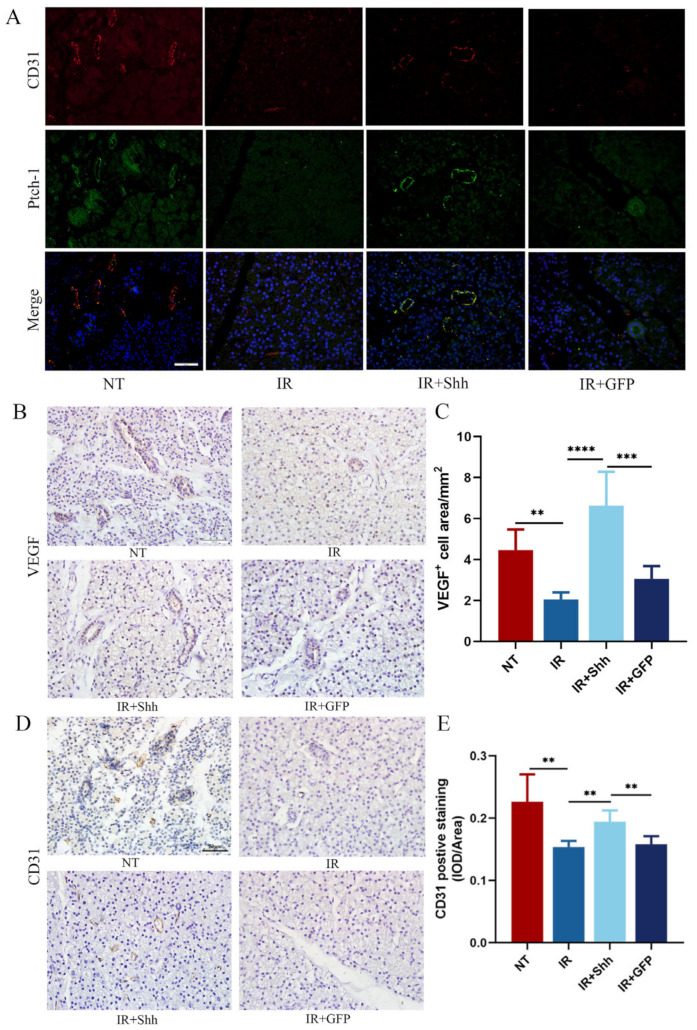
Shh transduction increased microvascular density in the parotid gland after IR. (**A**) The expression of PTCH1 and CD31 in parotid glands harvested 20 weeks after IR was assessed by double immunofluorescence staining. (**B**,**C**) IHC showed a significant decrease in the expression level of VEGF in the parotid glands of miniature pigs after IR. However, Shh gene therapy markedly increased VEGF secretion and effectively preserved vascular function after injury. The ratio of the VEGF-positive area in each visual field was calculated Scale bar, 50 μm; *n* = 5. (**D**,**E**) The expression of CD31 serves as a reliable marker for identifying endothelial cells. Immunohistochemical analysis revealed a significant reduction in microvascular density within the parotid gland following irradiation, while Shh gene therapy exhibited the potential to mitigate this alteration. The mean optical density of CD31^+^ cells in each field of view was calculated and subjected to statistical analysis. Scale bar, 50 μm; *n* = 5. The data are shown as the mean ± standard error of the mean (SEM); ns (not significant), ** (*p* < 0.01), *** (*p* < 0.001), and **** (*p* < 0.0001) indicate significant differences.

**Figure 3 antioxidants-13-00904-f003:**
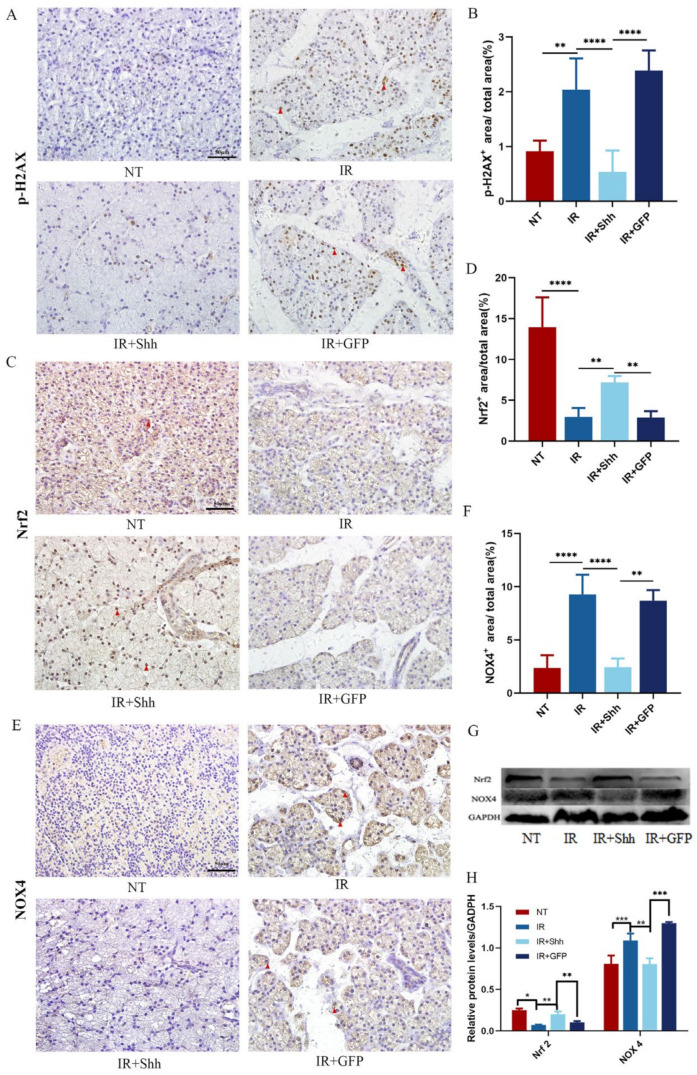
Shh gene delivery regulates oxidative stress levels by upregulating Nrf2 and downregulating NOX4. (**A**,**B**) Parotid tissues collected 5 weeks after IR were subjected to IHC to determine p-H2AX expression, and the ratio of the p-H2AX-positive area to the total area was statistically analyzed. The analysis revealed that Shh gene transduction inhibited the expression of p-H2AX (*p* < 0.0001) and reduced IR-induced DNA damage in parotid gland tissues. (**C**–**H**) The results of the IHC, Western blot, and quantification of Nrf2 and NOX4 in parotid tissues 5 weeks after IR indicated that Shh gene delivery activated Nrf2 (*p* < 0.01) and decreased NOX4 expression (*p* < 0.01). The data are shown as the mean ± standard error of the mean (SEM); ns (not significant), * (*p* < 0.05), ** (*p* < 0.01), *** (*p* < 0.001), and **** (*p* < 0.0001) indicate significant differences.

**Figure 4 antioxidants-13-00904-f004:**
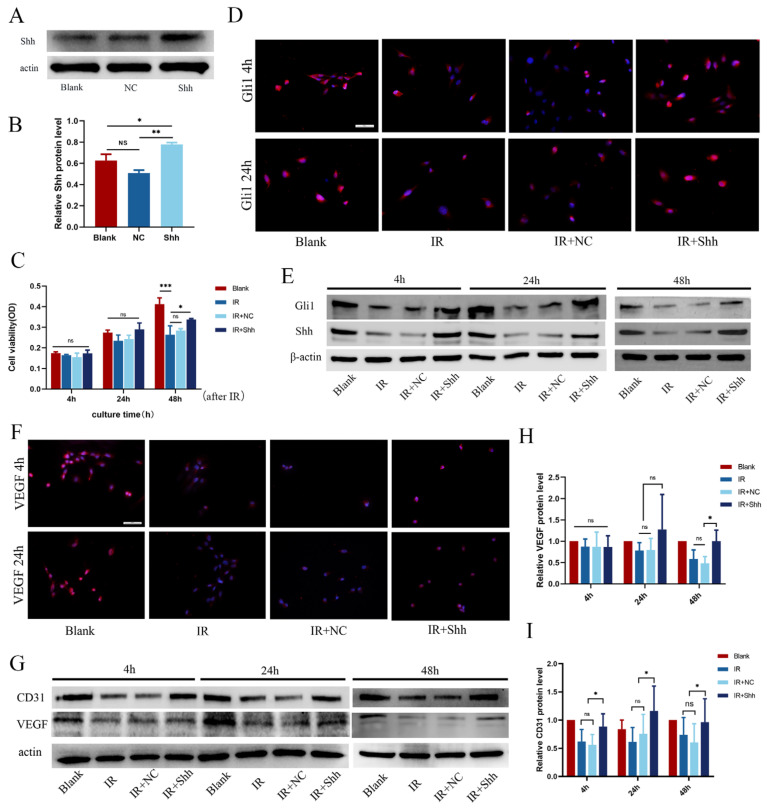
Shh gene overexpression activates the Hh signaling pathway and enhances vascular function in endothelial cells. (**A**,**B**) The protein expression level of Shh in the HUVECs in each group was detected by Western blotting, *n* = 3. (**C**) A CCK-8 assay revealed that Shh gene overexpression alleviated the inhibitory effect of IR on endothelial cell proliferation. *n* = 3, * *p* < 0.05. (**D**,**E**) Western blot and immunofluorescence staining results revealed that the Hh signaling pathway was significantly inhibited at 4 h, 24 h, and 48 h after IR. Furthermore, overexpression of the Shh gene was observed to activate the Hh signaling pathway and upregulate Gli1 expression. (**F**) IF staining revealed that compared with that in the blank group, VEGF expression in the IR and IR+NC groups was significantly lower at 4 h and 48 h after IR, whereas VEGF expression in the IR+Shh group was greater than that in the control group. Scale = 50 μm. (**G**–**I**) Western blot results and subsequent quantitative analysis revealed that overexpression of the Shh gene promoted the expression of CD31 and VEGF and restored vascular function in endothelial cells; *n* = 3, * *p* < 0.05. The data are shown as the mean ± standard error of the mean (SEM); ns (not significant), * (*p* < 0.05), ** (*p* < 0.01) and *** (*p* < 0.001) indicate significant differences.

**Figure 5 antioxidants-13-00904-f005:**
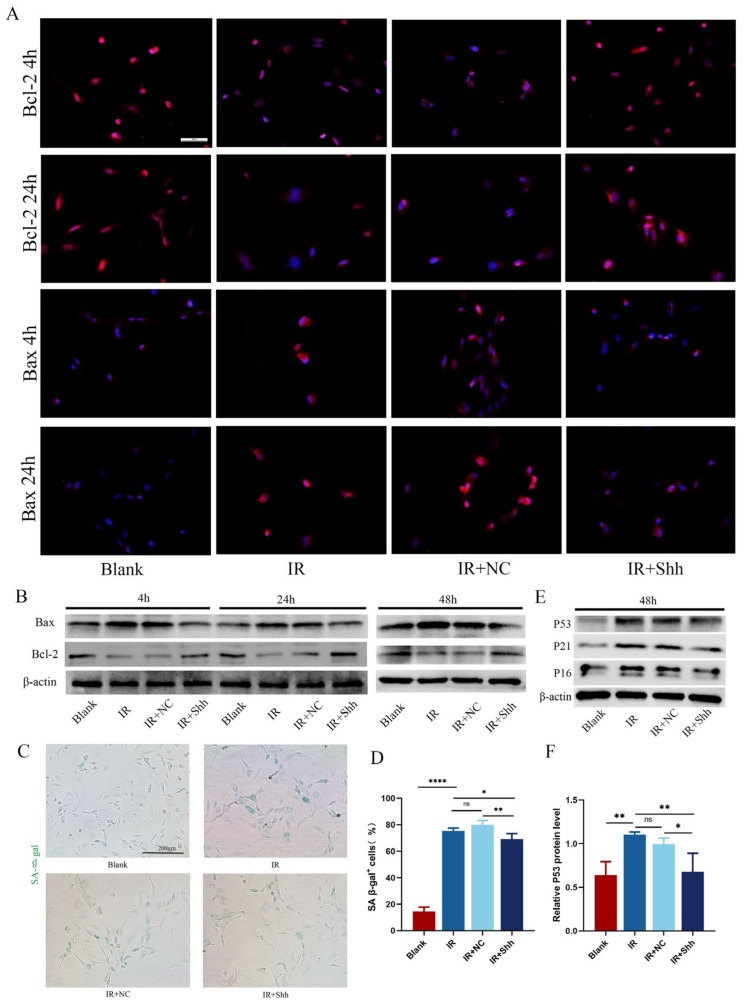
Activation of the Hh signaling pathway protects endothelial cells by regulating apoptotic signaling. (**A**,**B**) The levels of Bcl-2 and Bax in HUVECs were detected by IF staining and Western blotting. At 4 h, 24 h, and 48 h after IR, Shh overexpression resulted in a reduction in apoptosis rates, as evidenced by an increase in Bcl-2 expression and a decrease in Bax expression. (**C**,**D**) The number of β-gal-positive cells in the IR and IR+NC groups was significantly greater than that in the blank group, and the number of β-gal-positive cells in the IR+Shh group was significantly lower. *n* = 5, * *p* < 0.05, scale = 200 μm. (**E**) Western blotting was used to detect the expression of p53, p21, and p16. (**F**) Quantitative analysis of p53 protein expression level, *n* = 3. The data are shown as the mean ± standard error of the mean (SEM); ns (not significant), * (*p* < 0.05), ** (*p* < 0.01), and **** (*p* < 0.0001) indicate significant differences.

**Figure 6 antioxidants-13-00904-f006:**
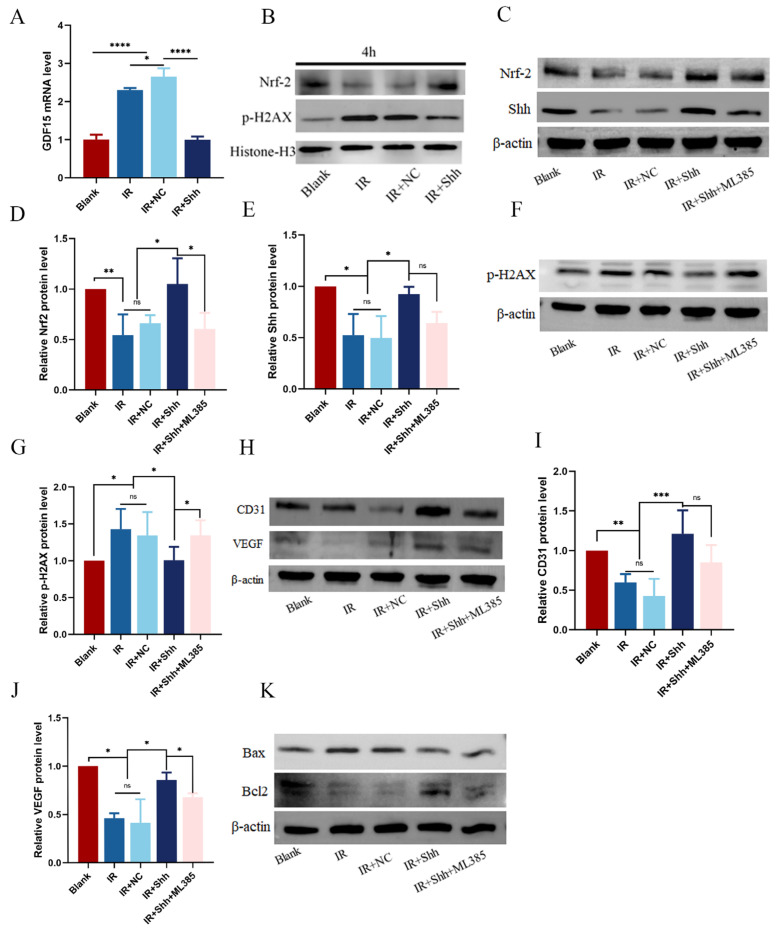
ML385 inhibits the antioxidative and vascular function-enhancing effects of Shh overexpression in HUVECs after IR. (**A**) The expression of GDF15 in HUVECs was analyzed by qRT-PCR. (**B**) p-H2AX expression was increased after IR, and overexpression of the Shh gene reduced p-H2AX production and activated Nrf2. (**C**–**E**) Treatment of HUVECs overexpressing Shh with ML385 before IR resulted in blockade of the Nrf2 pathway, while the expression level of Shh remained unaltered. (**F**,**G**) Western blot and quantitative analysis revealed that p-H2AX expression and DNA damage were increased following the inhibition of the Nrf2 pathway. (**H**–**J**) After activation of the Hh pathway, the promotion of VEGF and CD31 expression was inhibited by ML385. (**K**) ML385 pretreatment promoted endothelial cell apoptosis after IR. The data are shown as the mean ± standard error of the mean (SEM); ns (not significant), * (*p* < 0.05), ** (*p* < 0.01), *** (*p* < 0.001), and **** (*p* < 0.0001) indicate significant differences.

**Figure 7 antioxidants-13-00904-f007:**
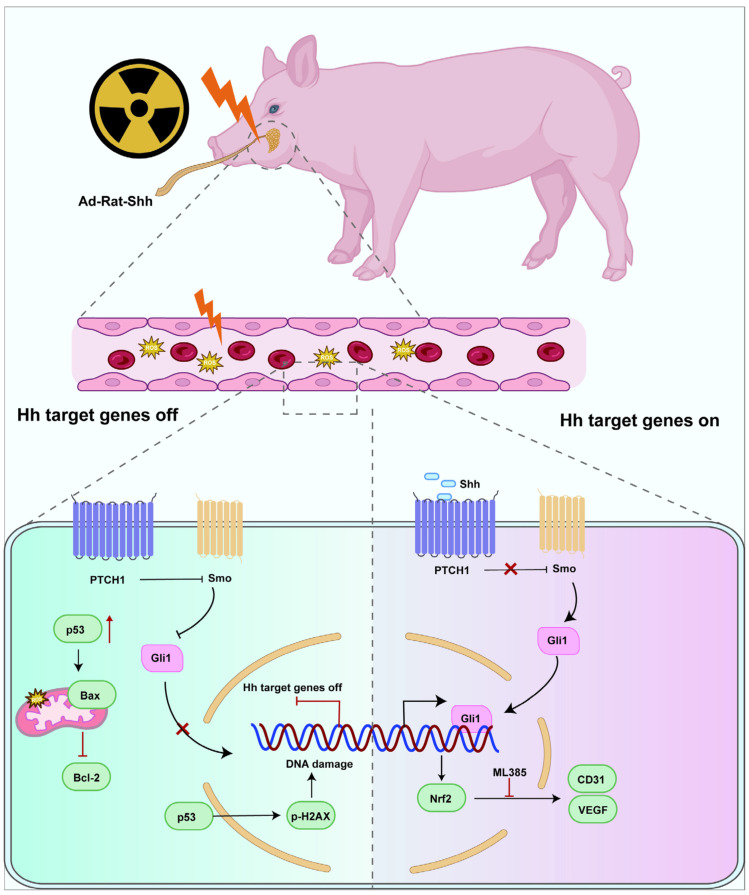
Graphical abstract. Intraglandular expression of the Shh gene efficiently attenuated IR-induced parotid gland injury in a miniature pig model. The antioxidative stress and microvascular protective effects of the Hh pathway are regulated by Nrf2.

## Data Availability

The data presented in this study is contained in the article.

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
