# Peer review of "Intragland Expression of the Shh Gene Alleviates Irradiation-Induced Salivary Gland Injury through Microvessel Protection and the Regulation of Oxidative Stress"

_antioxidants, 2024, doi:10.3390/antiox13080904_

Round 1

Reviewer 1 Report

The novelty of the presented study should be more highlighted. Please consider a published paper:

Hai B, Zhao Q, Deveau MA, Liu F. Delivery of Sonic Hedgehog Gene Repressed Irradiation-induced Cellular Senescence in Salivary Glands by Promoting DNA Repair and Reducing Oxidative Stress. Theranostics. 2018 Jan 13;8(4):1159-1167. doi: 10.7150/thno.23373. PMID: 29464006; PMCID: PMC5817117.

Some parts of the manuscript are hard to read. English language correction is recommended.

Specific comments:

The quality of some western blots is not satisfactory. The authors are asked to provide WB data of better quality.

Oxidative stress should be studied more directly. Please consider more accurate markers of oxidative stress. Please note that p-H2AX is a marker of DNA damage response (DDR). The same comment is true for the evaluation of cellular senescence. More markers are needed.

Limitations of the study should be highlighted.

The novelty of the presented study should be more highlighted. Please consider a published paper:

Hai B, Zhao Q, Deveau MA, Liu F. Delivery of Sonic Hedgehog Gene Repressed Irradiation-induced Cellular Senescence in Salivary Glands by Promoting DNA Repair and Reducing Oxidative Stress. Theranostics. 2018 Jan 13;8(4):1159-1167. doi: 10.7150/thno.23373. PMID: 29464006; PMCID: PMC5817117.

Some parts of the manuscript are hard to read. English language correction is recommended.

Specific comments:

The quality of some western blots is not satisfactory. The authors are asked to provide WB data of better quality.

Oxidative stress should be studied more directly. Please consider more accurate markers of oxidative stress. Please note that p-H2AX is a marker of DNA damage response (DDR). The same comment is true for the evaluation of cellular senescence. More markers are needed.

Limitations of the study should be highlighted.

Reviewer 2 Report

Xerostomia is a frequent and debilitating complication of head and neck cancer radiotherapy. Understanding its mechanisms may generate useful translational information. The present paper contributes to this issue. 

Results are convincing and illustrations are of good quality and appropriate. 

One. Explain all abbreviations when used first, e. g. Shh (L12), Hh and VEGF (L 18), etc. IR + NC (L 232) needs explanation: is NC the same as IR + GFP?

Two. Terminology. Should be uniform and describe the data. For example in section 3.4 and Figure 4 legend the terms “angiogenesis” , “vasogenic function” and “vascular function” are used probably meaning the same activity of endothelial cells, that is not tested. 

Three. Typos and syntaxis: used  used (L79); Elekata (L92); gas pedal (L93); SMOsi (L380)

Four. Provide further technical details and/or references about: measurement of salivary flow rate (L98); characterization of HUVEC (L104); dilated in the catheter (L174); delivery of Ad-Rat Shh (L95);  CCK-8 first mentioned in section 3.4 (L231); ML385 (L275); previous mouse studies (L425 & 456). 

Five. Figure legends need to describe not interpret the data; all symbols and abbreviations need explanation. In detail: 1B, what are the numerical units on the X-axis? 1C, add time. 1 D, E, G and I, describe changes in legend and label pictures. 2 B and D, describe changes in legend and label pictures. 2 C and E, explain numerical values in the Y-axis; replace “activation of the Hh pathway “ by “delivery of Ad-Rat Shh”.  . 3 B, D, F and H, explain numerical values in the Y-axis. 3 A, C, E label positive structure types; scale bars are different, is E (NT) at the same power as E (IR)? 4 A-B, the Western blot shows protein levels not gene expression. 4C, x-axis =?. 

Six. Discussion: A graphical abstract may greatly facilitate understanding of the molecular pathways involved in Shh modulation of ionizing radiation damage. 

Seven. Conclusion: L475, are clinical data available?

Round 2

Reviewer 1 Report

The authors have tried to improve the paper. The manuscript can be accepted for publication.

The authors have tried to improve the paper. The manuscript can be accepted for publication.